# Categorized and Correlated Multiple-Choice Questions: A Tool for Assessing Comprehensive Physics Knowledge of Students

**Shabnam Siddiqui**

Department of Chemistry and Physics, Louisiana State University Shreveport, Shreveport, LA 71115, USA; shabnam.siddiqui@lsus.edu

**Abstract:** An efficacious assessment tool is as necessary for improving physics education as are innovative and effective methods of teaching physics. Most tests focus on evaluating students knowledge in specific areas such as conceptual understating, quantitative, and analytical problem-solving skills. Testing students' critical thinking has remained a difficult task. Further, testing students comprehensive knowledge is even more challenging. We present here a new assessment tool with the acronym "Categorized and Correlated Multiple Choice Questions" (CCMCQs) for evaluating the comprehensive physics knowledge level of students. This tool consists of correlated questions posed in three different sub-categories of physics. Those sub-categories are: (i) Conceptual understanding, (ii) Critical thinking, and (iii) Quantitative understanding. The questions are structured to first pose a conceptual question which is then correlated with a critical thinking question and that critical thinking question is further correlated with a quantitative question. Thus, all three questions are correlated with each other, and such correlated questions aid the student and teacher alike to identify learning deficiency more accurately, and guide students to self-correct their knowledge of physics by providing appropriate direction. Further, we discuss the outcomes of a one semester study on CCMCQs using data obtained from an introductory physics course.

**Keywords:** physics education; multiple choice questions; critical thinking; comprehensive knowledge; assessment tool; correlated questions

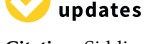

## 1. Introduction

Physics education research [1–3] has advanced significantly in the last several decades primarily in two areas. Several researchers have focused on identifying students' misconceptions, preconceptions, and common-sense conceptions in a physics classroom [4–7]. Some researchers have focused their research on understanding problem solving techniques and strategies [8,9] of the students. Primarily, misconceptions hinder students' understanding of the material presented to them in a classroom. Therefore, eliminating misconceptions from students' minds is necessary to any effective strategy for the teaching of physics. At the same time, teachers must help students to apply their conceptual and factual knowledge in a systematic way to solve practical problems [10–12]. The physics education research focused in these areas have resulted in the development of several active learning methods [13,14] that have provided an alternative to existing lecture style methods. These methods include Cooperative Groups [15], Socratic Dialogue Inducing Labs [16], Microcomputer-Based Labs [17], Interactive Demonstrations [18], Peer Instruction [6], Inquiry based tutorials [19], Studio Physics [20] and Integrated e-Learning [21]. These innovative methods of teaching have provided effective alternatives for the teaching of physics and im-proved engagement between students, peers, and teachers. Several studies have demonstrated that these methods are superior to the lecture style method of teaching. Also, for an evaluation of students' knowledge level of physics, researchers have developed effective testing methods such as: the Mechanics Baseline Tests [22], the Force Concept Inventory (FCI) [23], Test of Understanding Graphs in Kinematics [24] (TUG-K),

and the Conceptual Survey of Electricity and Magnetism [25], and several research-based assessment instruments [26,27] and Integrated e-learning [21]. All these tests are multiple-choice questions (MCQs) based tests. MCQs are therefore the most widely used assessment tool. Using MCQs-based tests researchers have been successful in identifying specific difficulties with or deficiencies in student understanding such as conceptual misconceptions, difficulty with the interpretation of graphs and quantitative deficiencies (difficulty with calculations using equations). Interestingly, such tests can point out the specific difficulty that students face in a physics course, but they do not provide complete information about the students' overall understanding of physics. MCQs based tests are not very reliable due to their several disadvantages which is discussed below in this article. Further, testing of students critical thinking in physics using MCQs has remained difficult because most of the textbooks offer only quantitative problems with some emphasis on conceptual problems. Critical thinking forms the basis of scientific thinking. It is not only important to teach critical thinking in a classroom. But evaluation of critical thinking is equally necessary in a 21st century education. Thus, there is a tremendous need to design assessment tools that evaluates students' wholesome knowledge of physics.

In this article, a new methodology for designing MCQs is presented. This methodology is a step towards improving of MCQs, and the development of tests that evaluates students' comprehensive knowledge of physics. It is based on categorization and correlation of physics knowledge. Using this methodology, a traditional textbook question can be broken down into several questions comprising of three categories. All the questions are logically connected or correlated. These correlated questions help the teacher and student alike to detect deficiencies and gaps in physics knowledge. To learn about the advantages of categorized and correlated multiple choice questions (CCMCQs), we conducted a study of CCMCQs based exams. In this study, we administered three exams over a semester during an introductory physics course and collected data from the exams. First, we performed item reliability test for each of the exams, and then performed data analysis using two approaches, Approach 1, and Approach 2. Using Approach 1, we analyzed data by considering in-dependent responses of the students. Whereas using Approach 2, we analyzed data by considering corelated responses of the students. Further, we compared these approaches to determine which approach offered the better approach to identify the deficiencies of the students. Further, several class discussions, and informal interviews were conducted with the students to verify the findings from the exams. These discussions helped significantly in uncovering students thinking processes. Since physics is most fundamental of all the sciences, it should be possible to apply this methodology to various other fields of science.

## 2. Background

MCQs-based tests are used to assess conceptual/qualitative and quantitative knowledge of physics and are used for both traditional and non-traditional classrooms. At several Universities, MCQs based tests are commonly used because they are easy to grade, low cost and most students favor their use for testing [28,29]. Particularly, for a large classroom, MCQs based tests reduces the teaching cost and for teachers, this process reduces their time load for grading [30,31]. It also permits them to focus their time more efficiently on the lecture content. Although, such a system provides instructors more time to plan and prepare for a lecture, it also reduces their understanding of the learning status and difficulties being confronted by their students. For traditional MCQs-based tests it is challenging for teachers to identify areas of weakness for the student since conceptual and quantitative questions are randomly presented or are presented under separate sections and there are no connections between the conceptual and quantitative questions. Such tests fail to indicate whether students lack a quantitative understanding is due to a lack of conceptual understanding or their conceptual and quantitative understanding of the material are independent of each other. Further, such tests provide no clues or guidance to students to answer a problem [32,33].

The impact of these assessment tools on students is that they rarely end up discussing their misconceptions and difficulties with the instructor or teaching assistant since they themselves are not certain what they are confused about. Secondly, the students continued uncertainty about their difficulties or misconceptions throughout the course tends to reduce their confidence level and it can often lead to a further reduction in the effectiveness of classroom lecture learning experience over time. Therefore, students are incentivized to seek alternative methods to improve their test score results. Such routes include: (i) memorization without understanding, (ii) chug and plug tricks for problem solving, (iii) guessing the right answer from the choices without any logic or understanding.

Thus, the overall disadvantages of MCQs based tests are that they [28–33]:

1.  foster a plug and chug style of answering
2.  encourage guessing strategies
3.  teach misinformation by disclosing wrong answers
4.  provide no guidance or clues to help answer a problem
5.  provide no feedback on deficiencies and misconceptions
6.  do not test critical thinking or reasoning

Alternately, an essay test which is also called as Free Response (FR) test where students are tasked with listings of all the steps for solving a problem or requires the students to write out the answer in words has only few of the above-mentioned disadvantages which are discussed later.

Given that MCQs have all these disadvantages, they cannot be effective in evaluating students' knowledge of physics even though they do not require as much effort/time on the behalf of the instructor(s). Below is an example [34] of a typical MCQ from an introductory physics course. The first question is a qualitative question and the second is a quantitative one.

---

**A sample of typical MCQs**

**Qualitative Question 1:** Bob throws a ball straight up releasing the ball at 1.5 m above the ground. Which of the following is a true statement?

(a)  The velocity of the ball increases constantly while rising and decreases constantly when falling.
(b)  The velocity of the ball decreases while rising and increases while falling.
(c)  The velocity of the ball increases while rising and decreases while falling.
(d)  The velocity of the ball decreases constantly while rising and increases constantly when falling.
(e)  The velocity of the ball remains same in both directions.
(f)  All the above.

This question's main goal is to evaluate conceptual understanding of how does the velocity of the ball change during its motion after being released by Bob.

**Quantitative question 2:** Bob throws a ball straight up at 20 m/s, releasing the ball at 1.5 m above the ground. Find the maximum height reached by the ball above the ground?

The maximum height of the ball is:

(a)  20.41 m
(b)  18.91 m
(c)  19.51 m
(d)  21.91 m

---

The main goal of the above question 2 is to test the student's quantitative understanding of basic physics. In a typical classroom on a physics test, both qualitative and quantitative questions are posed to the students. Usually, the questions are uncorrelated and are posed in different sections of the test. e.g., a conceptual section and a quantitative section. However, conceptual understanding is correlated with a quantitative understanding of physics. Therefore, posing uncorrelated questions in a test does not provide any feedback on students' overall learning deficiencies. In the above MCQ example, the stu-

dents who answered question 1 correctly, may still not be able to correctly answer question 2. The students who may answer question 2 correctly, may answer question 1 incorrectly. Thus, students may learn through their scores whether they have scored high or low in the conceptual section or the quantitative section. However, they will probably not be able to understand what their conceptual misunderstandings are which would prevent them from finding the correct answer to the question. For example, if a student answered question 1 correctly but answered question 2 incorrectly, then it will be more difficult for the student to understand what went wrong with his/her mathematical understanding that made him/her to choose the incorrect answer. Similarly, if a student answered question 2 correctly, and question 1 incorrectly, then it would be hard to understand why a student who has a sufficient mathematical understanding of the physics did not have sufficient conceptual understanding to answer question 1. In this example, both questions are correlated, thus it is more evident that the conceptual and quantitative/mathematical understandings are correlated. However, in a typical test conceptual questions are not correlated with quantitative/mathematical questions, therefore it is more challenging to determine a students' areas of difficulty. In some traditional classrooms, only quantitative/mathematical questions are included on physics tests. For example, to answer question 2 above, most students would not know where to begin or what to do to determine the steps to even understand how to resolve the problem. They are not conscious about their thinking processes. Some of the students may know how to apply the formula, which is simply "plug and chug" to find the answer to the question. They are not aware of the concepts involved in answering such a question. While others may try their luck by simply guessing.

In both an active learning-based classroom or in a traditional classroom, testing is mainly conducted with MCQs. It is assumed that by extensive practice with similar problems, students will develop sound conceptual understanding and reasoning ability. The students who fail to answer such questions do not benefit from feedback on their deficiencies. There is no scope for self-assessment and self-improvement. This is a "black box" approach, during which if students provide the correct answer, they gain a point, but are unlikely to be aware of their capability and skills. If they are incorrect in their answer, they do not know the reason or what is lacking in their knowledge of physics. The only feedback that a student receive is an overall score, and there is no explanation to uncover their deficiencies. The teachers also lack information about a students' deficiencies or how to help them. Thus, MCQs-based testing is not very effective in assessing students' overall knowledge and problem-solving strategies nor in assisting the teaching process to remedy deficiencies in their student's learning.

On the other hand, the advantages of MCQs are that they can be easily applied to any class size and are particularly helpful for large size classrooms. They are also easy to grade and require minimum test preparation. It is also clear that the convenience of these MCQs based tests for both students and teachers will provide a strong disincentive for their replacement by essay or subjective testing. Therefore, the question remains: Is there an alternative method for designing MCQs which can overcome all of the above-mentioned disadvantages and improve the learning outcomes in a physics classroom?

To improve MCQs based testing, researchers at the University of Colorado at Boulder [34] developed a new coupled multiple-choice response (CMR) test on the topic of electrostatics for an upper-division physics course. For this CMR's test, questions were designed to test student reasoning for a particular MCQ on a test. Their work was based on an already existing Colorado Upper-Division Electrostatics (CUE) diagnostics test, a Free Response (FR) test. It was designed as a FR test to gain understanding of a students' reasoning needed to craft MCQs. However, grading the CUE posed many challenges for graders. Therefore, an alternative test method was needed to overcome such issues. Thus, researchers at Colorado Boulder came up with a new CMR format for the CUE. They implemented this CMR CUE in several classrooms and studied their performance. Their research showed that the new CMR CUE works equally well for the FR format for gaining knowledge about students' reasoning and is also a viable tool for large scale implemen-

tation. However, further studies on this method are indicated to establish reliability and validity as an independent instrument. In another study by Lewdandowski [35], a set of CMR's were developed for upper-division physics labs. They also concluded that such tests are an improvement as compared to MCQs and far preferable for physics courses. Self et al. [36] reported CMR's usage for an introductory course on physics (mechanics). All these studies concluded that CMR's offered substantial improvement over standard MCQs for physics learning and student retention.

## 3. Methodology

Generally, in physics education, physics knowledge is described into three categories [34,37], (i) Factual knowledge, (ii) Conceptual knowledge and (iii) Procedural knowledge. Factual knowledge can be described as the knowledge of specific events and situation. Most of such knowledge is based on our own experience or gained through authority. Conceptual knowledge can be described as knowledge of physical principles, and knowledge that provides unified understanding of physical principles or knowing why. Procedural knowledge can be described as knowledge of how to apply factual and conceptual knowledge for problem solving. For example, students may have assumed (incorrectly) that heavier objects fall faster than lighter objects in all circumstances. However, in a physics class, students learn that all objects fall with the same acceleration under the assumption of the absence of air resistance. With such knowledge students' understanding of factual knowledge broadens and they learn that their factual knowledge is true under certain circumstances. Which is, in the presence of air resistance, a lighter object experiences more resistance to its motion than a heavier one and thus moves slowly compared to a heavier object. This broadened factual knowledge becomes conceptual knowledge and allows students to provide explanation for their experience (knowing why). Further, if a student may be asked to apply such conceptual knowledge to explain what would happen to an object's velocity when it is thrown upwards in the air (assuming absence of air resistance). Most of the students will have a hard time answering this question. Some of them would say that while moving up it slows down and while moving down it speeds up, which is an item of factual knowledge. However, if acceleration is constant then why does it decelerates in one direction and accelerates in the other? In my experience, I have found physics students have a really hard time in applying their conceptual knowledge to such situations because answering this question requires advanced reasoning and interconnections between two concepts i.e., velocity and acceleration. Such difficulty arises due to a lack of critical reasoning in understanding of the topic [38,39]. Critical thinking is least taught in a physics classroom [40,41]. Teachers have focused on imparting conceptual and procedural knowledge as separate pieces to the students. Thus, it is very important to connect the two pieces using critical thinking. So that critical thinking can be included as one of the subcategories of the physics knowledge.

In this work, physics knowledge is described in three general categories, conceptual understanding [23,25], critical thinking [38,39] and quantitative understanding (procedural knowledge) [11,15,24]. All three knowledge categories are intertwined or correlated. It is evident that without a solid conceptual understanding of the subject, critical thinking and quantitative understanding cannot be developed effectively. Without critical thinking, students cannot know how to apply conceptual knowledge for problem solving.

This method is described in more detail below in a flow chart format (Figure 1).

According to experts, critical thinking [38,39] can be described as a subset of three mental processes, reasoning, making judgements and decisions, and problem solving. As described in Table 1, critical thinking in physics can be ascribed to logical connections between different concepts (reasoning), detecting inconsistencies in reasoning (be able to decide what to believe based upon such reasoning), and systematic problem solving. In physics education, the main emphasis is on the teaching of conceptual knowledge and quantitative and analytical approaches to solve problems. Critical thinking is the least taught component of the mental processes in the physics classrooms. It is assumed that

students will develop critical thinking skills through extensive practice. Thus, teaching of critical thinking has remained challenging and is often mostly ignored. The interconnection between several concepts is taught randomly only through quantitative problem solving, and rarely discussed with the students. This gives rise to gaps in understanding of key concepts in physics. Most traditional textbooks offer only quantitative problems with some emphasis on the conceptual problems. Therefore, there is a tremendous need to design problems that not only test conceptual understanding and quantitative problem solving but also test the critical thinking of the students. Below, we discuss how to design problems in all three categories using a traditional problem from a physics textbook. By solving such problems students can gain sound understanding of the concepts of all the above mentioned three categories. This helps them to overcome gaps in understanding concepts and gain broader knowledge of physics. Further, tests should be designed to detect inconsistencies in the understanding of physics based upon these three categories of knowledge.

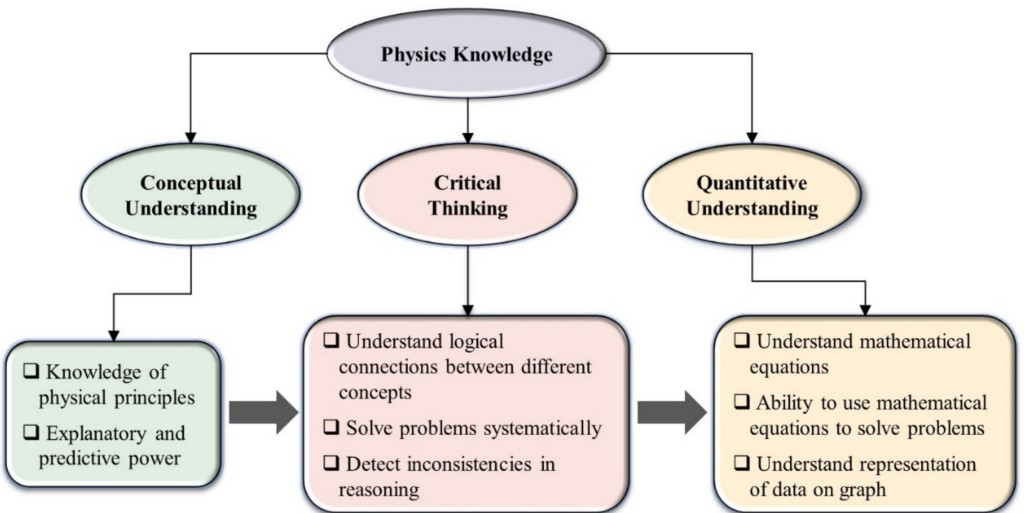

**Figure 1.** A flow chart describing physics knowledge based on three categories as conceptual understanding, critical thinking, and quantitative understanding.

### 3.1. Construction of Categorized and Correlated Multiple Choice Questions

An ideal MCQs-based test must test students conceptual understanding, critical thinking, and quantitative understanding. It should also provide clues to help solve a problem and provide the student with feedback about the subject areas that they need to improve their understanding of. In this article, a new approach for assessing students' conceptual knowledge is presented which includes critical thinking and quantitative understanding of physics. This approach has been assigned the acronym: Categorized and Correlated Multiple Choice Questions (CCMCQs). In this approach, a typical MCQ is broken down into different sets of MCQs, where each of them tests different categories of knowledge, such as conceptual understanding, critical thinking, quantitative understanding, and application of their overall knowledge to an actual relevant scenario. All the questions in different categories are correlated and provide a step-by-step process for learning and evaluation. Here the word 'correlated' means logical connection between the questions. Applying this approach, a students' knowledge can be assessed for each category. Also, it guides them to be able to solve problems systematically and provides them with automatic feedback on the areas requiring improvement. If the students fail to answer a conceptual question, they learn through the test that they are lacking in conceptual understanding, or if they fail in critical thinking, they learn through the test that they have a deficiency in critical thinking. Since all the questions are correlated (logically connected), it provides tightly correlated feedback on students' knowledge of physics. Thus, this assessment tool not

only assists students to understand the areas that require improvement but also helps the teacher to identify those areas as well. Further, the grading can be designed in such a way that the students can gain partial credits for each category, thereby reducing their chances to score very low or high by guesswork. The grading system provides clear discrimination between those students who do not understand the subject matter well and therefore do not score well and those who do score well because they understand the subject matter. It also minimizes the incentive for guesswork and offers guidance for solving a problem through a step-by-step process. Although, this tool provides an improvement over a MCQs-based test, it is required that such method must be verified through classrooms test to find out whether such a method is better or like free response questions-based tests in assessing students' depth of knowledge and creativity. Nevertheless, it is an improvement over existing MCQs based test and offers many of the same efficiency benefits of other MCQ-based tests, especially for large class sizes. An example of a CCMCQs-based correlated set of questions is listed below which illustrates the effectiveness of this approach as compared to traditional MCQs using the same example listed above. Below, we have taken a traditional question [34] and broken it down into four categories of correlated questions.

**Question:** Bob throws a ball straight up at 20 m/s, releasing the ball at 1.5 m above the ground. Find the maximum height reached by the ball. Assume no air resistance.
The above question is broken down into four subcategories as follows:

---

**Category 1: Conceptual Understanding**

Question Goal: To test conceptual understanding of the velocity of the ball.

**Question 1:** The force applied by Bob to throw the ball up causes the ball to
(a)     accelerate in the upward direction
(b)     gain initial velocity only
(c)     decelerate in the upward direction
(d)     gain initial velocity and acceleration
**Correct Answer: (b)**

**Question 2:** Which of the following is a true statement describing the ball's velocity for the entire motion?
(a)     The velocity of the ball constantly decreases by $-9.8$ m/s$^2$ while rising, becomes zero at maximum height, and constantly increases by $-9.8$ m/s$^2$ while falling down.
(b)     The velocity of the ball increases while rising and decreases while falling.
(c)     The velocity of the ball constantly decreases every second by 9.8 m/s while rising, becomes zero at a maximum height and constantly increases every second by 9.8 m/s while falling.
(d)     The velocity of the ball remains constant for the entire motion.
(e)     The velocity of the ball increases constantly by 9.8 m/s$^2$ while rising, becomes zero at maximum height and constantly decreases by 9.8 m/s$^2$ while falling.
**Correct Answer: (c)**

---

**Category 2: Critical Thinking**

Questions Goal: To test critical thinking (to test reasoning and detect inconsistencies between different concepts). In question 3 the main goal is to test whether a student can use reasoning to apply knowledge about the velocity to the concept of acceleration. In question 4, the main goal is to test whether a student has correct mental picture of motion of the ball and can identify correct known variables from the word problem. Here, g = 9.8 m/s$^2$.

**Question 3:** Which one of the following statements is true regarding acceleration of the ball?
(a)     the ball accelerates with a constant value of '+g' while rising because upward is a positive direction and decelerates with a constant value of '−g' when it falls because downward is a negative direction.
(b)     The ball accelerates with a constant value of '+g' while rising because both velocity and acceleration of the ball points in the same direction and decelerates with a value of '−g' when it falls because both velocity and acceleration points in opposite direction.

(c) The ball decelerates with a constant value of '$-g$' while rising because the velocity of the ball decreases constantly in the upward direction and accelerates with a constant value of '$-g$' when it falls because velocity increases constantly in the downward direction.

(d) The ball accelerates in both directions with a constant value of '$+g$' because gravitational acceleration is always constant and positive.

(e) The ball accelerates in both directions with a constant value of '$-g$' because gravitational acceleration always points in the downward direction.

**Correct Answer: (c)**

It is evident that the above three questions are correlated. If a student does not possess a correct conceptual understanding of the velocity in questions 1 and 2 then he/she may answer question 3 incorrectly. Even if the student ends up answering question 2 correctly by a lucky guess, then we know that since the student does not possess a correct understanding of question 1 then he/she may have guessed their answer for question 2 and 3. Using this information, a teacher can provide feedback to a student for improving their understanding of the concept of velocity.

**Question 4:** Which one of the following diagrams represent the ball's motion after release into the air with all the known variables applicable to the ball.

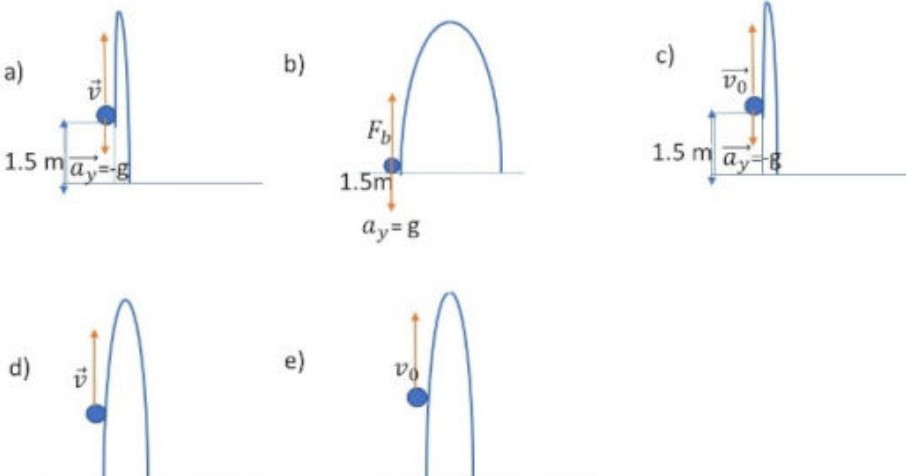

**Correct Answer: (c)**

Here questions 3 and 4 are correlated.

**Category 3: Quantitative Understanding**

Question Goal: Application of the equations of motion

The questions below (question 5 & 6) are correlated with questions 3 & 4.

**Question 5:** The known variables are:

(a) Initial position, initial velocity, final velocity at maximum height, and acceleration
(b) Initial position, final position, and initial velocity
(c) Initial velocity, final velocity at maximum height
(d) Initial position, initial velocity, and acceleration

**Correct Answer: (a)**

**Question 6:** The unknown variables are:

(a) The maximum height of the ball above the initial position and final velocity when it hits the ground
(b) The final velocity when it hits the ground and its acceleration
(c) Its maximum height
(d) Its maximum height and acceleration

**Correct Answer: (a)**

**Questions 7:** The correct equation for finding the maximum height is:

(a)   $v^2 = v_0^2 - 2g(y - y_0)$
(b)   $v^2 = v_0^2 + 2g(y - y_0)$
(c)   $y - y_0 = v_0 t + \frac{1}{2}gt^2$
(d)   $y - y_0 = v_0 t - \frac{1}{2}gt^2$
(e)   $v = v_0 + gt$

**Correct Answer: (a)**

**Question 8:** The maximum height reached by the ball from the ground is:

(a)   20.41 m
(b)   18.91 m
(c)   19.51 m
(d)   21.91 m

**Correct answer: (d)**

For the above category 3, it is clear that if a student answers the first three questions incorrectly or one of them incorrectly then the chances of them answering the fourth question (question 8) correctly are considerably lower. For example, if a student answers question 6 incorrectly then we know that a student is having trouble in identifying the unknown variables in the problem and will likely have difficulty in determining the correct equation for solving the problem. Also, if a student answer all the questions correctly except question 6 then it is likely that the student has guessed answers or has answered the questions by applying an unstructured strategy. Thus, both teacher and student can identify and learn from the deficiency. At the same time, by practicing the above questions, a student can be directed into a systematic learning strategy for solving problems.

Question Goal: Quantitative representation of the data of the ball's motion on a graph

**Question 9:** The speed vs time graph that represents the speed of the ball during its entire motion is:

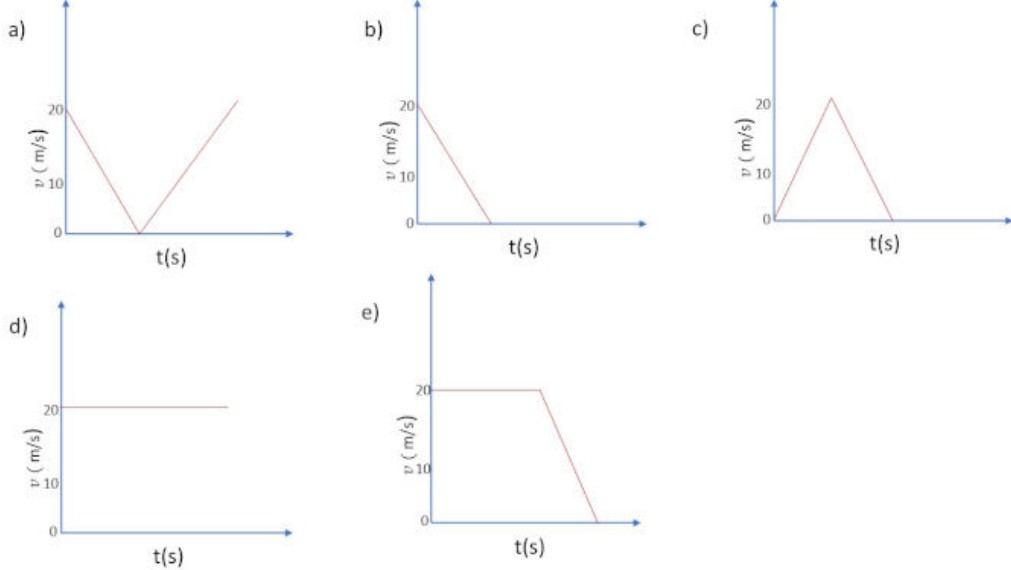

**Correct Answer: (a)**

**Category 4: Application of the Physics Knowledge to a Real-World Scenario**

**Question 10:** Consider that you and your friend are standing on top of a building which is 10 floors high, and you both need to determine the height of the building. What will you do to determine the height of the building approximately? You both have cell phones and a ball. Assuming that the wind is negligible on that day.

(a)   You will ask your friend to stand at ground zero of the building, you will then drop the ball and use your cell phone to record the time the ball takes to hit the ground, and then applying the following equation $(-h = -\frac{1}{2}gt^2)$, you can then calculate the height of the building.

<div style="border:1px solid">

(b)   You will ask your friend to stand at ground zero of the building, you will then drop the ball and use your cell phone to record the initial velocity of the ball, time taken by the ball to hit the ground and then applying the following equation ($h = v_0 t + \frac{1}{2} g t^2$), you can calculate the height of the building.

(c)   You will ask your friend to stand at ground zero of the building, you will then drop the ball and use your cell phone to determine the initial acceleration 'a' of the ball, time taken by the ball to hit the ground and then using the following equation ($-h = v_0 t - \frac{1}{2} a t^2$), you can calculate the height of the building.

(d)   You will ask your friend to stand at ground zero of the building, you will then drop the ball and use your cell phone to record the initial velocity of the ball, time taken by the ball to hit the ground and then applying the following equation ($-h = v_0 t - \frac{1}{2} g t^2$), you can calculate the height of the building.

**Correct Answer: (a)**

</div>

As is evident from the questions listed above, if a student fails to answer the question from category 1 correctly then the student is most likely to answer all the remaining questions inconsistently correct as well. In this case, both the teacher and the student will be able to learn from this deficiency and the teacher can direct the student appropriately to improve his/her knowledge of physics. In this way, CCMCQs allows partial credit along and automatic feedback on deficiencies. Below is a table that provides comparison between Free response (FR), MCQs, and CCMCQs based tests.

**Table 1.** A comparison between FR test, MCQs based test and CCMCQs based test.

| Category | FR Test | MCQs Test | CCMCQs Test |
|---|---|---|---|
| Credit | Credit is based on the number of steps completed correctly. A student may obtain anywhere between partial to full credit. | Credit is based on a correct or incorrect answer. Full credit for correct answer and no credit for an incorrect one. No partial credit. | Credit is based on answering the number of correlated MC questions correctly. A student may obtain anywhere between partial to full credit. |
| Deficiencies/ difficulties | Instructor/Student learns about the deficiencies of the students during the grading process. | Instructor/Student is unaware of their difficulties and misconceptions. | Instructor/students learns about the deficiencies during the grading process. |
| Human error | Instructor can make mistakes during the grading process which can affect a student's grade. i.e., there is scope for human error. | Grading is accomplished by machine, and there is little or no scope for human error. | Grading is accomplished by a machine, and there is little or no scope for human error. |
| Guessing or random chance strategies | A student cannot obtain full credit by guessing the answer. A student must complete all the correct steps for solving the problem to arrive at the correct answer. | A student can guess the correct answer even though he/she does not have an appropriate level of knowledge of the subject. | Since, the question is divided into several categories, a student may obtain partial credit for guessing. This method also helps to identify whether a student has guessed an answer. |
| Step-by-step learning | None | None | Aid in systematic learning |

### 3.2. Administration of CCMCQs in Three Exams

During a one semester study, data from CCMCQs based three exams was collected at Louisiana State University at Shreveport (LSUS) for an introductory physics course for non-physics majors (trigonometry and algebra-based physics). The sample size for the data collection comprised of 74 samples for three exams (24–25 students per exam) in a semester. The exams were designed to evaluate students' performance in two categories: (i) Conceptual understanding of a physics topic, and (ii) Quantitative understanding of the physics topic which requires the use of correct equations and the calculation of a correct value of the physical quantity involved. These categories are correlated and from which it follows that to have complete understanding of the physics concept, students must have a sound understanding of the physics concept involved and apply the correct equation with correct intermediate algebraic steps that lead to a calculation of correct value of the

physical quantity. Thus, if a student does well in both categories, then one can conclude that he/she must have gained complete understanding of the concept. The main reason for choosing two categories instead of the three categories as discussed above is that since this is our first attempt as a preliminary trial we need to start at the most basic level. As this research continues, the question strategy will be expanded to include all three categories listed above.

Applying this approach, it can be determined whether students have a sound understanding of the physics concepts and if they apply them appropriately to solve a problem or are they just plugging and chugging to solve a problem without a sound understanding of the concept involved. In these exams we have deliberately minimized asking questions in the critical thinking category. Only in exam 2 was a conceptual question posed that required interconnection between two of the physics concepts. Below, we discuss the logical basis for the construction of three different exams.

### 3.2.1. Construction of CCMCQs for Exam 1

At the beginning of the semester, the students were introduced to topics such as the use of trigonometry in physics, applications of vectors for solving physics problems, and kinematics in one dimension that includes freely falling bodies. A set of four CCMCQs were presented on the topic of freely falling bodies in Exam 1. The topic of freely falling bodies is discussed with no introduction of forces. Out of four questions, two were conceptual questions, and the remaining two were quantitative questions. The first two questions were designed to test students understanding of acceleration due to gravity when an object is thrown upwards. It was shown in the class that in the absence of air resistance when an object is thrown upwards, it decelerates while rising at a constant rate of $9.8 \text{ m/s}^2$ (g), finally comes to a halt momentarily at some height, then reverses direction and accelerates with the same constant rate while falling. The main student misconception is that since acceleration remains negative in both directions, the object must decelerate in both directions. If it accelerates while falling, then its acceleration must be positive. They do not realize that since acceleration is a vector, and the object is falling, its acceleration is negative even though it is accelerating in the downward direction. Therefore, acceleration in both directions remain negative but for different reasons. So, all four questions were designed to test whether students correctly understood this concept. The remaining two questions were designed to test whether students can apply the correct knowledge of acceleration due to gravity to systematically solve a problem in the calculation of a physical parameter such as maximum height reached by the object when it is thrown upwards. Below is a sample question from the first exam.

**Question.** At the start of a football game, a referee tosses a coin upward with an initial velocity of $v_0 = 5 \text{ m/s}$. While rising in the air, the coin

(a)   accelerates constantly because its velocity keeps increasing every second by 9.8 m/s.
(b)   decelerates constantly because its velocity keeps decreasing every second by 9.8 m/s.
(c)   velocity remains constant
(d)   accelerates with a constant value of $9.8 \text{ m/s}^2$ and thus velocity remains constant

**Correct Answer: (b)**

### 3.2.2. Construction of CCMCQs for Exam 2

CCMCQs were presented on the topic of friction. The categorized questions were designed to test, (i) a conceptual understanding of friction + critical thinking, (ii) the application of the correct equation for the calculation of the amount of force required to move a body from the state of rest, and (iii) the determination of the numerical value of the maximum static frictional force that can act on a body. The students had major misconceptions regarding static frictional force. A majority of the students thought that when a body is at rest, it is because a static frictional force is preventing it from moving. When asked, whether there is a static frictional force acting on a body when no external force is applied to it, their response is that a static frictional force acts on a body when it is in

a state of rest even when no external force is applied to the body. This misconception is very common with physics majors as well. Even when taught that there is no static frictional force acting on a body when no external force is applied to the body, a majority of students refuses to believe this fact. They have such a strong misconception that static frictional force automatically acts on a body and causes it to be in a state of rest. It seems that students have been inculcated with this misconception because of the way this concept is taught at high school or in other courses, and because the name 'static frictional force' suggests to them that it is a force that causes an object to be in a state of rest (static) irrespective of an external force. It seems that their difficulty arises because of their inability to connect the Newton's first law of motion to the concept of static frictional force and perhaps a poor choice of the identifier for the force. Thus, they would seem to lack in critical thinking. A sample question from Exam 3 is given below.

---

**Question:** Consider a block lying on the floor. The block is stationary because

(a)    A static frictional force acts horizontally (x-axis) on the block and keeps it stationary.
(b)    No static frictional force is acting on the block, thus the net force on the block along x-axis is zero. But the net force along y-axis is not zero.
(c)    No static frictional force is acting on the block, thus the net force on the block along x-axis is zero, and the net force on the block along y-axis is also zero.
(d)    No forces are acting on the block.

**Correct Answer: (c)**

---

### 3.2.3. Construction of CCMCQs for Exam 3

CCMCQs were presented to the students on the topic of uniform circular motion. The categorized questions were designed to test, (i) their conceptual understanding of circular motion, (ii) their choice of the correct equations for calculating velocity and centripetal acceleration and the unknown quantity from the equation for centripetal force, and (iii) their determination of the numerical value of the unknown physical quantity which required intermediate algebraic steps. Below is a sample question from exam 3.

---

**Question:** A ball is moving uniformly in a circle. Which of the following is a true statement?

(a)    The direction of the ball's velocity changes continuously, and its magnitude remains constant for all time. Whereas the acceleration of ball remains constant.
(b)    The ball's velocity is constant half of the time, and its acceleration is constant for all time.
(c)    The ball's velocity changes for all time. The acceleration of the ball is constant for all time.
(d)    The ball's velocity changes constantly in direction only, and its magnitude remains constant for all times. Whereas the magnitude of acceleration of the ball is constant, and its direction continuously changes.

**Correct Answer: (d)**

---

### 3.3. Evaluation of CCMCQs for the Exams

In physics education, several statistical techniques [42,43] are used to analyze and interpret data obtained from multiple choice tests. In this work, we have conducted item analysis which comprises of three measures, item difficulty (P), discrimination index (D) and point biserial coefficient ($r_{pbi}$). Item difficulty is a measure of easiness of an item in a test, and discrimination index measures how powerful an item is in distinguishing high-achieving students from low achieving students and point biserial coefficient measures individual item reliability. Since our main goal was to understand the easiness or hardness of the questions posed in the three exams, we first measured item difficulty for each question, and later measured point biserial coefficient to determine reliability of each question. The three items on which we conducted these measures are, (i) Item 1 comprised of the first two conceptual questions, (ii) Item 2 was a question on usage of the correct equation, and (iii) Item 3 was a question of the correct numerical value of a physical quantity. The item difficulty was defined as a measure of the relative degree of easiness of the item and was calculated as a proportion of correct responses as given below,

$$P = \frac{N_1}{N} \qquad (1)$$

Here $N_1$ is the number of correct responses to an item, and $N$ is the total number of students. Ideally, items with difficulty level around 0.5 have highest reliability because questions that are extremely difficulty or very easy do not discriminate between students. Practically, item value ranges between 0.3 to 0.9 are acceptable. If items with extremely low or high difficulty is detected, then one may consider revising such items.

Secondly, to determine the reliability of each item in the exam, we measured point biserial coefficient for each of the tests in the exams. Here, a test means a set of four CCMCQs questions in the exam. This coefficient is a measure of individual item reliability and is defined as the correlation between item scores and total scores of a test,

$$r_{pbi} = \frac{\overline{X}_1 - \overline{X}_0}{\sigma_x} \sqrt{P(1-P)} \qquad (2)$$

Here, $\overline{X}_1$ is the average total score for those who correctly answer an item, $\overline{X}_0$ is the average total score for those who incorrectly answer the item, $\sigma_x$ is the standard deviation of total scores, and P is the difficulty index for this item. A reliable item should be consistent with the rest of the test, so a high correlation between the item score and the total score is expected. A satisfactory point biserial coefficient is $r_{pbi} \geq 0.2$. Data was collected from students' responses to all these three tests, and standard deviation to students' scores, difficulty index, and the biserial coefficient were calculated for each test of these three exams. Below, Table 2 present these values calculated for each item for all the exams.

**Table 2.** Difficulty index (P), and biserial coefficient calculated values for each item for the three exams.

|  | Item 1: P, $r_{pbi}$ | Item 2: P, $r_{pbi}$ | Item 3: P, $r_{pbi}$ | $\sigma_x$ |
|---|---|---|---|---|
| **Exam 1** | 0.48, 0.70 | 0.64, 0.40 | 0.88, 0.40 | 0.57 |
| **Exam 2** | 0.28, 0.62 | 0.80, 0.28 | 0.48, 1.33 | 0.57 |
| **Exam 3** | 0.42, 1.1 | 0.78, 0.91 | 0.78, 1.0 | 0.75 |

The difficulty index (P) values for three items for each exam lies within the acceptable range. For exams 1 and 3, the P value is highly reliable for item 1 which is close to 0.5. Thus, this clearly demonstrates that the exam questions were appropriate as far as their difficulty measure is concerned. Further, the values for the biserial coefficient for each item in the three exams also lay within the acceptable range. Therefore, this table confirms that all the questions in the exam were reliable and demonstrated direct correlation between their item and total scores.

## 4. Results and Discussions

*Data Analysis*

The data obtained from three exams was analyzed using two statistical approaches, Approach 1, and Approach 2. These approaches are described below.

*Approach 1:* This approach is most applied in physics education research for studying student responses during a test. It is comprised of studying students' responses as independent responses without any correlations between questions. Here, we have collected data for incorrect responses of the students for sets of four questions in three exams.

This data illustrates the percentages of incorrect student responses as shown in Figure 2. This data shows that students had major difficulties in answering question 1 in exam 2 and exam 3, and least difficulty in answering question 4 of exam 1 and exam 2. Since question 1 in exam 2 is a conceptual question on static friction. This suggest that students had major difficulty in understanding this concept. Similarly, second major difficulty of the students was in answering question 1 of exam 3. This also was a conceptual question on the concept of circular motion. This suggest that students had major difficulties in

answering conceptual questions on these two topics. The least difficulty of the students was in answering question 4 in all three exams. Which was numerical values of physical quantities such as amount of static frictional force and centripetal force. This data does provide information regarding where majority of the students are having difficulty. But it does not provide complete information about overall difficulties of the students. For example, it does not provide any information regarding how many students who answered conceptual question correctly ended up answering numerical question incorrectly and vice-versa, and how many students are struggling, and not gaining any better understanding of physics. From a more detailed review, it was discovered that some of the students who answered the conceptual question correctly, answered question 2 incorrectly and some of the students who answered question 1 incorrectly answered question 2 & 3 correctly. Further, it is almost impossible to gain any information concerning how many students just guessed the correct answers.

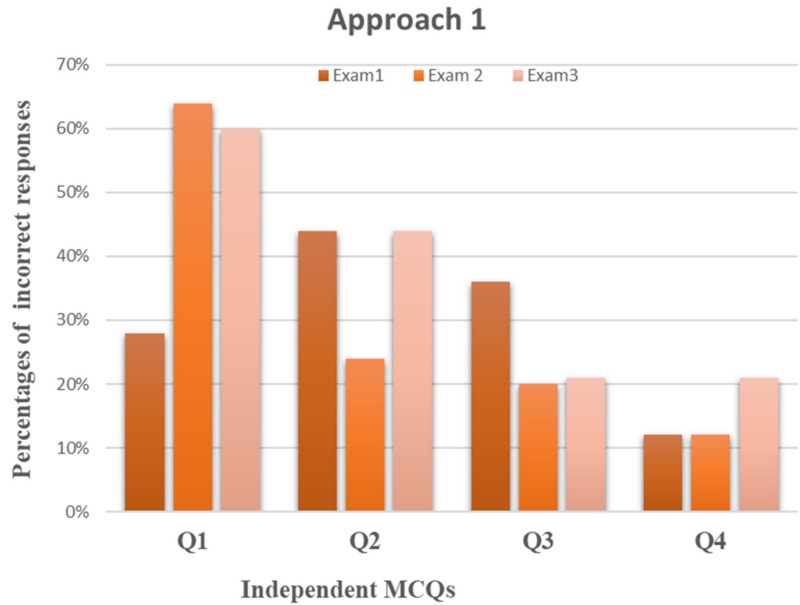

**Figure 2.** Percentages of the students' incorrect responses for MCQs (Q1, Q2, Q3 and Q4) from three exams.

This analysis demonstrates that Approach 1 does not provide complete knowledge of the students' difficulties. It just points out some specific difficulties. For example, the data suggested that the conceptual understanding of physics is most difficult for the students. Also, students somehow can choose a correct equation for solving a problem and are even better at arriving at correct numerical value. This suggests further investigation of possible combinations of correct and incorrect responses of the students.

*Approach 2:* For this approach, correlated responses of the students were examined. All four responses were combined into a set of three responses. Thus, CCC (Correct, Correct, Correct) means that students answered all three items correctly. These students appeared to have had a sound understanding of the concept and knew how to solve the problem systematically. CCI (Correct, Correct, Incorrect) means that they answered first two conceptual questions correctly and answered one of the quantitative questions on equation correctly but answered incorrectly the numerical value question, which implied that students had a very sound understanding of the concept and had a partial quantitative understanding of how to solve the problem systematically, and correctly. CII (Correct, Incorrect, Incorrect) means correct response to only first item on conceptual questions, which implied they had only some understanding of the concept but did not know which equation to apply for solving the problem. CIC (Correct, Incorrect, Correct) means a correct

response to the first two conceptual questions and correct response to last quantitative question, which means they lacked complete understanding of the concept, and did not know which equation to apply to solve the problem but somehow arrived at the correct numerical value. ICC (Incorrect, Correct, Correct) means incorrect response to both or any of the two conceptual questions which means students had either a complete lack of understanding of the concept or a partial understanding of the concept, but knew which equation to use to solve the problem and arrived at a correct numerical value of the parameter. III (Incorrect, Incorrect, Incorrect) means all incorrect responses to all the questions. IIC (Incorrect, Incorrect, Correct) means incorrect conceptual responses, incorrect usage of equations but correct choice for the numerical value of the parameter. This suggests a random response or guesswork. ICI (Incorrect, Correct, Incorrect) means only one correct response to a question on an equation and remaining all incorrect responses, which implied either they had only partial understanding of the concept or guessed the answer for this question. Such answer combinations and the number of students corresponding to each combination for each exam are listed in the table below (Table 3).

**Table 3.** A combination of correct and incorrect responses of the students from three exams.

| Correct (C) and Incorrect (I) Combinations of Answers | Exam 1 No. of Students | Exam 2 No. of Students | Exam 3 No. of Students |
|:---:|:---:|:---:|:---:|
| CCC | 5 | 4 | 9 |
| CCI | 2 | 0 | 0 |
| CII | 0 | 0 | 0 |
| CIC | 5 | 3 | 1 |
| ICC | 9 | 15 | 7 |
| III | 1 | 1 | 3 |
| IIC | 3 | 1 | 2 |
| ICI | 0 | 1 | 2 |

Based on this understanding, we categorized students responses into five categories as, (i) a sound overall understanding of the concept where students answered all four questions correctly (A), (ii) a sound understanding of the concept and correct usage of the equation but had difficulty in calculating the correct numerical value (N), (iii) difficulty in understanding the concept but correct use of the equation to solve the problem, and to calculate correct numerical value (C), (iv) Guesswork, and random correct responses (G), (v) answered all questions incorrectly (F). Since, the questions were correlated, this approach may provide a better understanding of students' deficiencies. A bar graph of students' responses in these five categories for all three exams is shown below (Figure 3).

The data does suggest some remarkable conclusions. For example, for exam 1, it indicated, (A) 20% of the class appeared to have had a sound understanding of the concept, and knew how to solve the problem systematically and arrive at a correct numerical value of the problem, (C) 36% of the class had difficulty in understanding the concept but can use the equation correctly to solve the problem, (N) 8% had a sound understanding of the concept, and knew which equation to use to solve the problem but had difficulty in calculating the correct numerical value, and (G) 20% + 12% = 32% of the class responses were random, and were just lucky in guessing the correct answers. Lastly, (F) only 4% of the class answered all questions incorrectly. The overall trend shows that students who had all correct responses for exam 3 increased by 18% relative to exam 1 & exam 2, and their conceptual understanding improved as well (as shown in C category for exam 3 in Figure 2), their guess work was reduced in frequency as they become more confident about their understanding of physics. Also, students who gave all incorrect responses slightly increased. As the semester progresses, the course material tends to become more complex. Physics requires interconnections between various concepts to be able to acquire a sound conceptual understanding and thus it requires more effort to learn. Although students

make progress in understanding physics concepts throughout the semester, the gradual increase in complexity of the course material tends to obscure the appearance of progress. In a one semester course, it is a challenge for a teacher to be able to help students overcome misconceptions and gain an improved understanding of the interconnections between the various concepts. This is where CCMCQs based tests can be of assistance to teachers and students alike to keep track of the challenges involved and to facilitate teaching strategy improvements based on these findings.

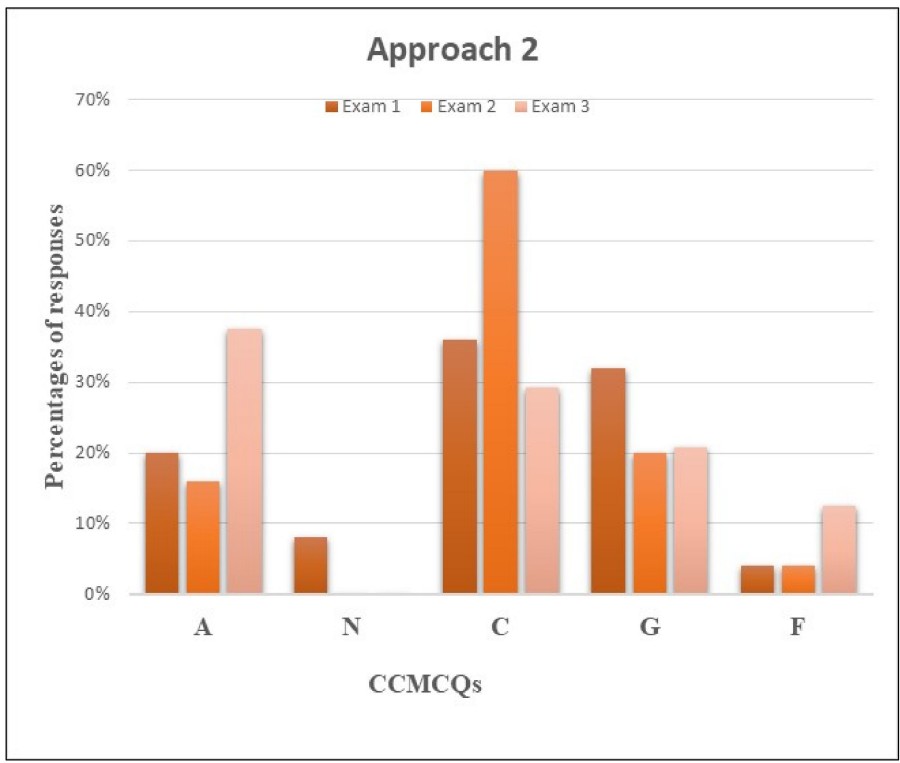

**Figure 3.** The percentages of the students' responses in five categories (A, N, C, G, and F) as correlated questions (CCMCQs) from the three exams.

To better understand the conclusions from the data, several informal interviews and discussions were conducted in the classroom to discuss the students' perceptions of the exams. During these discussions, all the possible incorrect answers were discussed. Some of the students provided their reasoning for choosing incorrect response. For example, the word 'static friction' suggests to them that it is the force that causes an object to be in a state of rest, and this misconception tended to direct them to choose an incorrect answer. While some students struggled to describe their thought processes but still confirmed that CCMCQs-based tests helped them to identify their difficulties. Most of the students had least difficulty in understanding the questions (or the language of the questions) posed in the tests which is clearly suggested by our item reliability test. Finally, an incorrect response is an incorrect response. It mainly points out two possibilities, either the student had a lack of understanding of the concept or the way the question was posed in the test caused confusion for the student and prompted them to choose an incorrect answer. The overall trend from the statistical data indicated the number of students who acquired full or partial understanding of the concepts. Such insights are impossible to obtain from a typical MCQs-based tests. For future studies, more formal student interviews will be conducted to gain a further detailed understanding of student thinking processes for choosing correct and incorrect responses, and critical thinking questions will be included in the tests.

## 5. Conclusions

This paper presents a new CCMCQs-based assessment tool and illustrates its application of CCMCQs to an introductory physics course. It also discusses several advantages of CCMCQs over traditional MCQs. Further, this work demonstrates the potential of such a tool to evaluate student knowledge of the physics through studies of the CCMCQs based exams conducted at LSUS. The outcome of this study suggests that this tool aids in measuring data on student performance in a test that provide a complete understanding of the student learning challenges as compared to a data obtained from MCQs based test. For this work, two statistical approaches were applied to analyze the data obtained from the three exams. Approach 1 was comprised of analysis of data for independent sets of correct and incorrect responses while Approach 2 was comprised of analysis of data for correlated correct and incorrect responses. Compared to Approach 1, Approach 2 provided more detailed information about student difficulties. The analysis of the data for these two approaches suggests that the major student learning difficulty was in understanding the interconnections between different concepts in physics (critical reasoning), and their least difficulty was in applying equations to obtain a correct numerical value. By studying several combinations of correct and incorrect responses for the correlated questions, the percentages of the students that possess a sound understanding of the physics concepts and solve the problem systematically was determined. In addition, the percentages of the students that lack a conceptual understanding but are effective at applying equations and arriving at the correct numerical value, the percentages of the students that just guess at the answers, and percentages of the students that are struggling with overall physics concepts was also determined. The data also suggests that if a student lack understanding of the physics concept in one category, then their responses for the other category will be inconsistent as well. The informal interviews and class discussions provided clearer understanding of students' incorrect responses and difficulties with specific aspect of the curriculum. This work suggests that CCMCQs can be an effective tool to identify students learning deficiencies more accurately and they aid the teacher in the choice of more effective strategies to address these deficiencies. It also aids the students to learn how to solve problems more systematically, and to self-correct their deficiencies. In addition, this tool is more accurate for grading than MCQs, since it allows partial credit for a response which is only possible with the FR-based tests. We hope that this tool can be developed with numerous studies for various physics courses and will aid in learning of physics.

## 6. Limitations and Future Studies

This work is a preliminary step towards the development of this methodology, and further research is continued for development of the approach as an assessment instrument. In this study, a control group is not used to compare the data, secondly formal interviews are needed to validate students' responses, and more emphasis on critical thinking questions is needed. In our future studies, we will incorporate these improvements.

**Funding:** This research was funded by Louisiana State University in Shreveport, and the grant number is 1234.

**Institutional Review Board Statement:** The study was conducted in accordance with the Declaration of Helsinki and approved by the Institutional Review Board of Office of Sponsored Research and Technology Transfer (LSUS Institutional Review Board) (protocol code LSUS IRB# 2021-00038 and the date of approval is 5 May 2022).

**Informed Consent Statement:** Informed consent was obtained from all subjects involved in the study.

**Acknowledgments:** I would like to thank Louisiana State University at Shreveport for providing Dean's Faculty Support Fund, and Ian Wylie for help with editing of the manuscript. Also, I would like to thank all the students whose data I have used for the research.

**Conflicts of Interest:** The author declares no conflict of interest.

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
